# Phyco-Synthesized Zinc Oxide Nanoparticles Using Marine Macroalgae, *Ulva fasciata* Delile, Characterization, Antibacterial Activity, Photocatalysis, and Tanning Wastewater Treatment

Amr Fouda [1,*], Ahmed M. Eid [1], Ayman Abdelkareem [1], Hanan A. Said [2], Ehab F. El-Belely [1], Dalal Hussien M. Alkhalifah [3], Khalid S. Alshallash [4] and Saad El-Din Hassan [1]

[1] Department of Botany and Microbiology, Faculty of Science, Al-Azhar University, Cairo 11884, Egypt; aeidmicrobiology@azhar.edu.eg (A.M.E.); aymanegybotany@gmail.com (A.A.); elbelely@azhar.edu.eg (E.F.E.-B.); saad.el-din.hassan@umontreal.ca (S.E.-D.H.)

[2] Botany Department, Faculty of Science, Fayoum University, Fayoum 63514, Egypt; hah01@fayoum.edu.eg

[3] Department of Biology, College of Science, Princess Nourah Bint Abdulrahman University, P.O. Box 84428, Riyadh 11671, Saudi Arabia; dhalkalifah@pnu.edu.sa

[4] College of Science and Humanities-Huraymila, Imam Mohammed Bin Saud Islamic University (IMSIU), Riyadh 11432, Saudi Arabia; ksalshallash@imamu.edu.sa

* Correspondence: amr_fh83@azhar.edu.eg; Tel.: +20-111-335-1244

**Abstract:** The aqueous extract of marine green macroalgae, *Ulva fasciata* Delile, was harnessed for the synthesis of zinc oxide nanoparticles (ZnO-NPs). The conversion to ZnO-NPs was characterized by color change, UV–vis spectroscopy, FT-IR, TEM, SEM-EDX, and XRD. Data showed the formation of spherical and crystalline ZnO-NPs with a size range of 3–33 nm. SEM-EDX revealed the presence of Zn and O in weight percentages of 45.3 and 31.62%, respectively. The phyco-synthesized ZnO-NPs exhibited an effective antibacterial activity against the pathogenic Gram-positive and Gram-negative bacteria. The bacterial clear zones ranged from $21.7 \pm 0.6$ to $14.7 \pm 0.6$ mm with MIC values of 50–6.25 $\mu g\ mL^{-1}$. The catalytic activity of our product was investigated in dark and visible light conditions, using the methylene blue (MB) dye. The maximum dye removal ($84.9 \pm 1.2\%$) was achieved after 140 min in the presence of 1.0 mg mL$^{-1}$ of our nanocatalyst under the visible light at a pH of 7 and a temperature of 35 °C. This percentage was decreased to $53.4 \pm 0.7\%$ under the dark conditions. This nanocatalyst showed a high reusability with a decreasing percentage of ~5.2% after six successive cycles. Under the optimum conditions, ZnO-NPs showed a high efficacy in decolorizing the tanning wastewater with a percentage of $96.1 \pm 1.7\%$. Moreover, the parameters of the COD, BOD, TSS, and conductivity were decreased with percentages of 88.8, 88.5, 96.9, and 91.5%, respectively. Moreover, nano-ZnO had a high efficacy in decreasing the content of the tanning wastewater Cr (VI) from $864.3 \pm 5.8$ to $57.3 \pm 4.1$ mg L$^{-1}$ with a removal percentage of 93.4%.

**Keywords:** green synthesis; *Ulva fasciata*; ZnO-NPs; pathogenic bacteria; methylene blue; chromium ions

## 1. Introduction

Nanoparticles (NPs) are defined as particles at the nanoscale (1–100 nm) and they are considered building blocks of nanotechnology [1]. Due to the beneficial and unique physical, chemical, and mechanical properties of NPs, they are used in various applications such as energy, cosmetics, medicines, catalysis, electronics, wastewater treatment, heavy metal adsorption, food, the textile industry, and agriculture [2–5]. Therefore, the global production of NPs has been receiving high attention and is estimated to be 260–309 metric tons [6], whereas the global consumption was in the range of 255–585 metric tons from 2014 to 2019 [1]. The production of NPs was accomplished by chemical and physical methods such as microwave, solvothermal, sol-gel, hydrothermal synthesis, chemical reduction,

laser ablation, γ-ray irradiation, the evaporation–condensation method, and flame spray pyrolysis [7]. Although these methods are widely used for the production of specific shapes and sizes, the adverse effects of these methods on the environment, researchers, and consumers, such as the toxic effects of the organic solvent involved in NP synthesis, the use of extreme conditions such as high pH, high temperature, and the production of toxic vapors and gases that have adverse effects on humans and the ecosystem, can retard their usage [7,8]. Green chemistry or the green synthesis of nanoparticles has been concerned with the development of an eco-friendly approach to manufacture the NPs. This approach promises to realize the increasing demands for the production of environmentally friendly NPs with a low cost, biocompatibility, easy scale-up, saving energy, and easy handling methods [9,10].

Marine macroalgae are classified based on their pigmentation to brown ((phaeophytes), green (chlorophytes), and red (rhodophytes)) and are characterized by renewable resources and are easy to be collected from the marine environment. Algal biomass (alive or dried) has received more attention as a "bio-nano-factory" because of its rapid growth, high capacity for metal and metal oxide accumulation and reduction, high biomass productivity, growth in the absence of chemicals or fertilizers, and the ability to harvest it several times per year [11,12]. Among green macroalgae, *Ulva fasciata* Delile belongs to the class Ulvophyceae and the family Ulvaceae, and it is widely distributed on the north coast of Egypt. The aqueous extract of these green macroalgae contains various bioactive molecules such as polysaccharides (Ulvan), amines, amides, proteins, pigments, terpenoids, alkaloids, and phenolic compounds that have a critical role in metal and metal oxide reduction and stabilization [13]. Therefore, the current study emphasizes the potential of *U. fasciata* as a low-cost and high-active metabolites content to fabricate Zinc oxide nanoparticles (ZnO-NPs) by a green approach. In a recent study, magnesium oxide (MgO) nanoparticles were formed by harnessing the metabolites produced by the brown algae *Cystoseira crinita* [14]. Various NPs such as Ag, Au, AgCl, CuO, and $Fe_3O_4$ were successfully fabricated by marine macroalgae [15].

The ZnO-NPs are important semiconductors due to their high bandgap, electron mobility, and visible transparency. They are also incorporated into a wide variety of applications such as optoelectronics, optics, the rubber industry, the textile industry, cosmetic and sunscreen products because of their high properties of UV adsorption, solar cells, concrete production, photocatalysis, biological sensing, gene delivery, and larvicidal, antibacterial, and antifungal agents [16–18]. Bacterial infections and the increasing antibiotic-resistant microbes are the main challenges, and a development of new bioactive compounds to overcome their negative effects on human and animal health is very important [19,20]. ZnO of nanoscale-size particles has significant effects as an antimicrobial agent against Gram-positive bacteria, Gram-negative bacteria, and uni- and multicellular fungi compared to the microscale-size particles. For instance, the ZnO-NPs synthesized by the aqueous extract of *Ulva lactuca* (UL-ZnO-NPs) and *Stoechospermum marginatum* (sm-ZnO-NPs) exhibited high antibacterial activity against *Staphylococcus aureus*, *Escherichia coli*, *Salmonella typhi*, and *Proteus vulgaris* and antifungal activity against *Aspergillus flavis*, *A. niger*, and *Fusarium oxysporum* more effectively than the metal precursor ($ZnSO_4$) and the algal aqueous extract [21]. This phenomenon could be attributed to the unique properties of ZnO of nanoscale-size particles, especially because of the large surface area relative to the small size [22]. The discharge of effluents from different industries, such as textile and leather tanning, without treatment causes massive harmful effects on the environment [23]. This is due to the highly toxic substances of these effluents, such as dyes and heavy metals [24,25]. Recently, the development of new semiconductors that are used as catalysts for remediation of these effluents are of global interest. The photocatalytic remediation of industrial effluents using NPs is considered a promising tool due to its simplicity, efficiency, rapid oxidation, cost-effectiveness, and the reduction in toxic by-products synthesis [1,26]. The photocatalytic reaction depends on the surface area of the catalyst, crystalline structure, size, shape, size distribution, bandgap, porosity, and the density of hydroxyl groups on

the catalyst surface [27]. Due to the superior properties of ZnO-NPs as a semiconductor, a plethora of studies were conducted to investigate their efficacy in the degradation and adsorption of different pollutants and heavy metals.

In the current study, the efficacy of the aqueous extract of marine macroalgae, *U. fasciata* Delile, in reducing zinc nitrate and forming ZnO-NPs was investigated. The phyco-synthesized ZnO-NPs were initially noticed by the color change of the reaction solution. Next, an extensive characterization was performed using UV–vis spectroscopy, Fourier transform infrared (FT-IR) spectroscopy, Transmission Electron Microscopy (TEM), Scanning Electron Microscopy–energy-dispersive X-ray (SEM-EDX), and X-ray diffraction (XRD). The activity of the developed ZnO-NPs as antibacterial agents against the pathogenic Gram-positive bacteria (*Staphylococcus aureus* and *Bacillus subtilis*) and the Gram-negative bacteria (*Escherichia coli* and *Pseudomonas aeruginosa*) was studied. The catalytic activity of phyco-synthesized ZnO-NPs to degrade the methylene blue (MB) as a model dye was assessed in dark and visible light conditions. We determined an optimum condition in which the effectiveness of ZnO-N *U. fasciata* Ps to improve the physicochemical parameters of tanning wastewater for the first time as well as to adsorb the chromium ions were investigated.

## 2. Results and Discussion

### 2.1. U. fasciata-Mediated Green Synthesis of ZnO-NPs

In the current study, the active metabolites involved in the aqueous extract of *U. fasciata* act as a biocatalyst for reducing zinc acetate to zinc oxide nanoparticles (ZnO-NPs). Carbohydrates, fats, vitamins, fatty acids, bioactive metabolites such as polyphenols, and pigments (carotenoids, phycobilin, and chlorophyll) are present in the aqueous extract of algae and act as reducing and stabilizing agents for forming a wide range of metal and metal oxides nanoparticles as reported previously [5,15]. Herein, the formation of a white color after mixing the algal aqueous extract with the metal precursor of $Zn(CH_3COO)_2 \cdot 2H_2O$ indicates the successful fabrication of ZnO-NPs. The obtained data are in agreement with the previous reports which indicated that the aqueous extract of *U. lactuca* has the efficacy to form white precipitate after mixing with zinc acetate which indicates the formation of ZnO-NPs [28]. Similarly, the aqueous extract of the seaweed *Hypnea musciformis* was used to form a white precipitate of $Zn(OH)_2$ after mixing with zinc nitrate that was calcinated at 450 °C for 2 h to form ZnO-NPs [29].

The fabrication of ZnO-NPs using an algal aqueous extract could be due to one of two possible mechanisms. First, the potential of biomolecules and active metabolites that exist in the aqueous extract to chelate the $Zn^{2+}$ to form a complex of $Zn(OH)_{2'}$ which is collected and calcinated at 300 °C for 2 h to form a white powder as ZnO-NPs in Equations (1) and (2) [18,30].

$$Zn(CH_3COO)_2 \cdot 2H_2O + \text{seaweed aqueous extract} \xrightarrow[\text{incubated overnight under dark conditions}]{\text{Stirring for 60 min at 50°C}} Zn(OH)_2 \quad (1)$$

$$Zn(OH)_2 \xrightarrow[\text{2 h}]{300°C} \text{ZnO-NPs} \quad (2)$$

The second mechanism might be through the reduction of $Zn^{2+}$ to form zinc metal ($Zn^0$) via the active metabolites present in the aqueous extract, followed by reacting with the dissolved oxygen to form the nuclei of the ZnO. Next, the as-formed ZnO is capped by some algal metabolites that may increase the stability of nanomaterials and prevent its aggregation [31]. Due to the promising activity of ZnO-NPs in various biomedical and biotechnological applications, it was synthesized using a green, cost-effective, and eco-friendly approach by a variety of biological entities [32–34].

*2.2. Characterizations*

2.2.1. UV–vis Spectroscopy

The intensity of the white color that formed after mixing the aqueous extract of *U. fasciata* with the zinc acetate solution was monitored by UV–vis spectroscopy at a wavelength in the range of 200–600 nm (Figure 1). It was shown that the UV–vis spectra of the algal aqueous extract have a maximum absorption peak at 440 nm that shifted to 330 nm after the formation of the ZnO-NPs. The observed maximum peak at 330 nm was matched with the bandgap absorption of nanocrystal ZnO as reported previously [28,32]. Similarly, the ZnO-NPs fabricated by the macroalgae *Sargassum muticum* showed a sharp absorption peak at a wavelength of 334 nm and attributed this band to the transition of electrons from the valence band (VB) to the conductance band (CB) in the ZnO sample [35]. Moreover, the maximum absorption peak of the ZnO-NPs synthesized by *U. lactuca* was observed at the wavelength of 325 nm [28]. Recently, the ZnO-NPs synthesized by the two seaweeds, *Stoechospermum marginatum* and *U. lactuca*, have a maximum absorption peak at the wavelength of 345 and 310 nm, respectively [21].

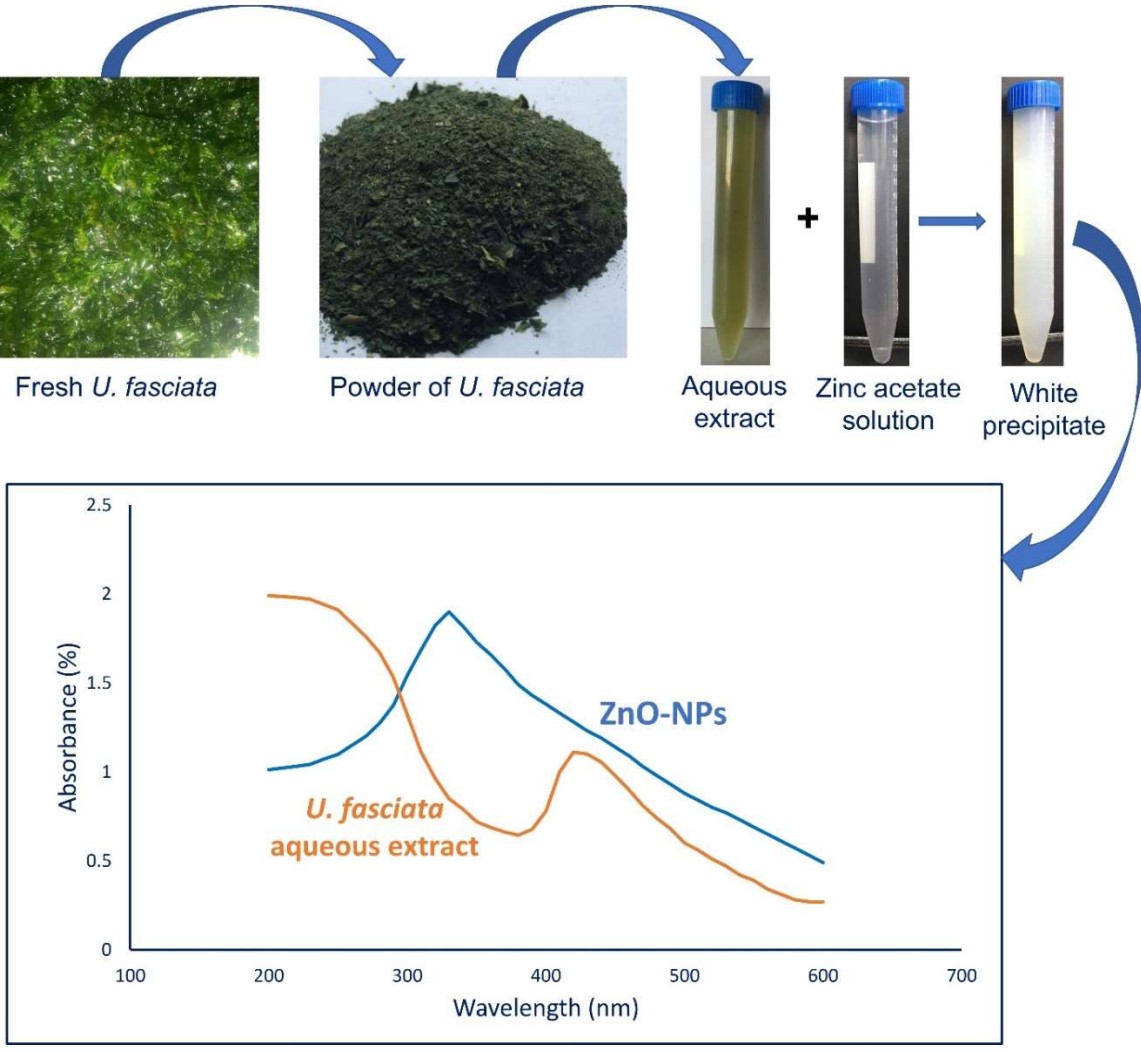

**Figure 1.** Phyco-synthesis of ZnO-NPs using aqueous extract of *U. fasciata* and monitoring the intensity of formed color using UV–vis spectroscopy at a wavelength of 200–600 nm.

2.2.2. Fourier Transform Infrared (FT-IR) Spectroscopy

The functional groups related to various active compounds that are present in the algal aqueous extract and their role in the reduction process as well as the capping and

stabilizing of ZnO were investigated by FT-IR. As shown, the aqueous extract of *U. fasciata* has five peaks at the wavenumbers 528, 1113, 1630, 2070, and 3450 cm$^{-1}$ (Figure 2). The peaks in the range of 950–1150 cm$^{-1}$ have been reported for typical ulvans polysaccharides secreted by *U. fasciata* [36]. The strong peak at 1630 cm$^{-1}$ is attributed to the C=O of polysaccharide moieties, whereas the broad peak at 2070 cm$^{-1}$ signifies the stretching N=C=S of isothiocyanate [37]. The strong broadness peak at 3450 cm$^{-1}$ corresponds to the stretching of O—H and N—H for amide A proteins [38]; this peak was shifted to 3430 cm$^{-1}$ in the ZnO-NPs. The broadness peak was formed due to the formation of intra- or intermolecular hydrogen bonds [39]. The FT-IR spectra of the ZnO-NPs showed a weak peak at 2990 cm$^{-1}$ which is related to the asymmetric and symmetric stretching CH$_2$ of carbohydrates and lipids [40] (Figure 2). The peak at 2429 cm$^{-1}$ is signified to CO$_2$ adsorption, whereas the shifting peak to 1640 cm$^{-1}$ can be related to the stretching C=N (amides) and C=O overlapped with bending N—H amines [41,42]. The appearance of a new, strong peak at 1380 cm$^{-1}$ is corresponding to the stretching C–N group of aromatic and aliphatic amines, whereas the medium peak at 1325 cm$^{-1}$ is signified to O–H bending [43]. Moreover, the medium peak at 835 cm$^{-1}$ is signified to the bending C=O of alkene. The successful formation of ZnO was confirmed due to the presence of peaks at 514 and 775 cm$^{-1}$ [44,45]. The previous peaks were overlapped with a band of C–S at an absorbance of 528 cm$^{-1}$ of the algal aqueous extract [14]. The FT-IR results revealed that the organic contents of *U. fasciata* such as proteins, amino acids, carbohydrates, and polysaccharides were involved in the reduction of zinc ions and the capping and stabilizing of the phyco-synthesized ZnO-NPs.

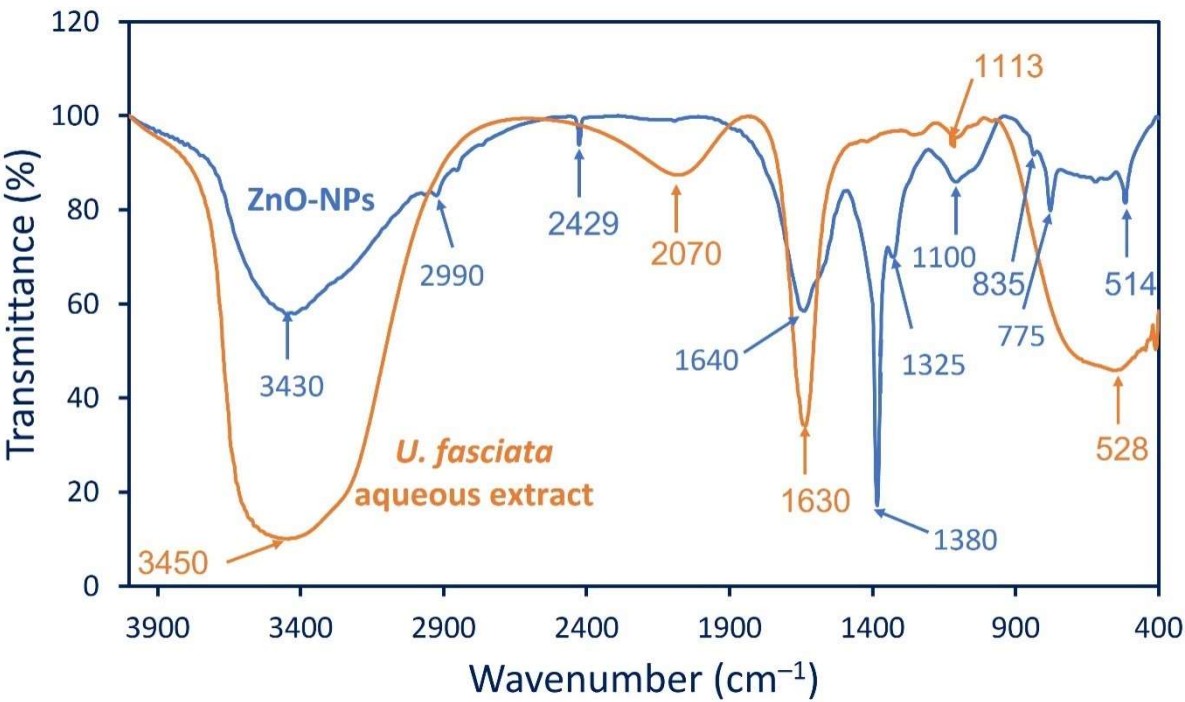

**Figure 2.** FT-IR analysis showed functional groups in aqueous extract of *U. fasciata* and their role in the green synthesis of ZnO-NPs.

### 2.2.3. Transmission Electron Microscopy (TEM)

The activity of nanoparticles depends on various parameters such as size, aggregation, shape, surface charge, and stability [46,47]. Therefore, it is important to investigate the morphological characteristics, especially the size, shape, and aggregation of phyco-synthesized ZnO-NPs. The TEM instrument is a useful technique for assessing these parameters. As shown, the synthesized ZnO-NPs were well-dispersed without aggregation and have a spherical shape (Figure 3A). Moreover, the size of the ZnO-NPs was in

the range of 3–33 nm with an average size of 10.62 $\pm$ 4.9 nm (Figure 3B). The activity of the ZnO-NPs differed according to the size and shape. For instance, the antimicrobial activity of the cuboidal shape of the ZnO-NPs synthesized by the leaf extract of *Aloe vera* was more effective against the pathogenic bacteria, *Escherichia coli, Bacillus subtilis*, and *Staphylococcus aureus*, than the hexagonal, spherical, and cylindrical shapes. Whereas the photocatalytic efficacy of the spherical followed by hexagonal shapes was more effective in the removal of methyl orange dye than the cuboidal and the cylindrical shapes [48]. Interestingly, the green-synthesized nanorod ZnO exhibited a high activity against the pathogenic Gram-positive and Gram-negative bacteria and in-vitro cytotoxicity compared with the hexagonal shape. Moreover, after loading on cotton fabrics, the nanorod shape was more safe than the hexagonal one [33]. Moreover, various sizes of ZnO-NPs (20, 40, and 140 nm) showed varied antimicrobial activity against *Streptococcus mutans, Enterococcus faecalis, Lactobacillus fermentum*, and *Candida albicans* based on their sizes [49]. As previously reported, the activity of the ZnO-NPs increases as their size decreases [8,46]. Hence, we predicted high activity of our ZnO-NPs product because of its relatively small size.

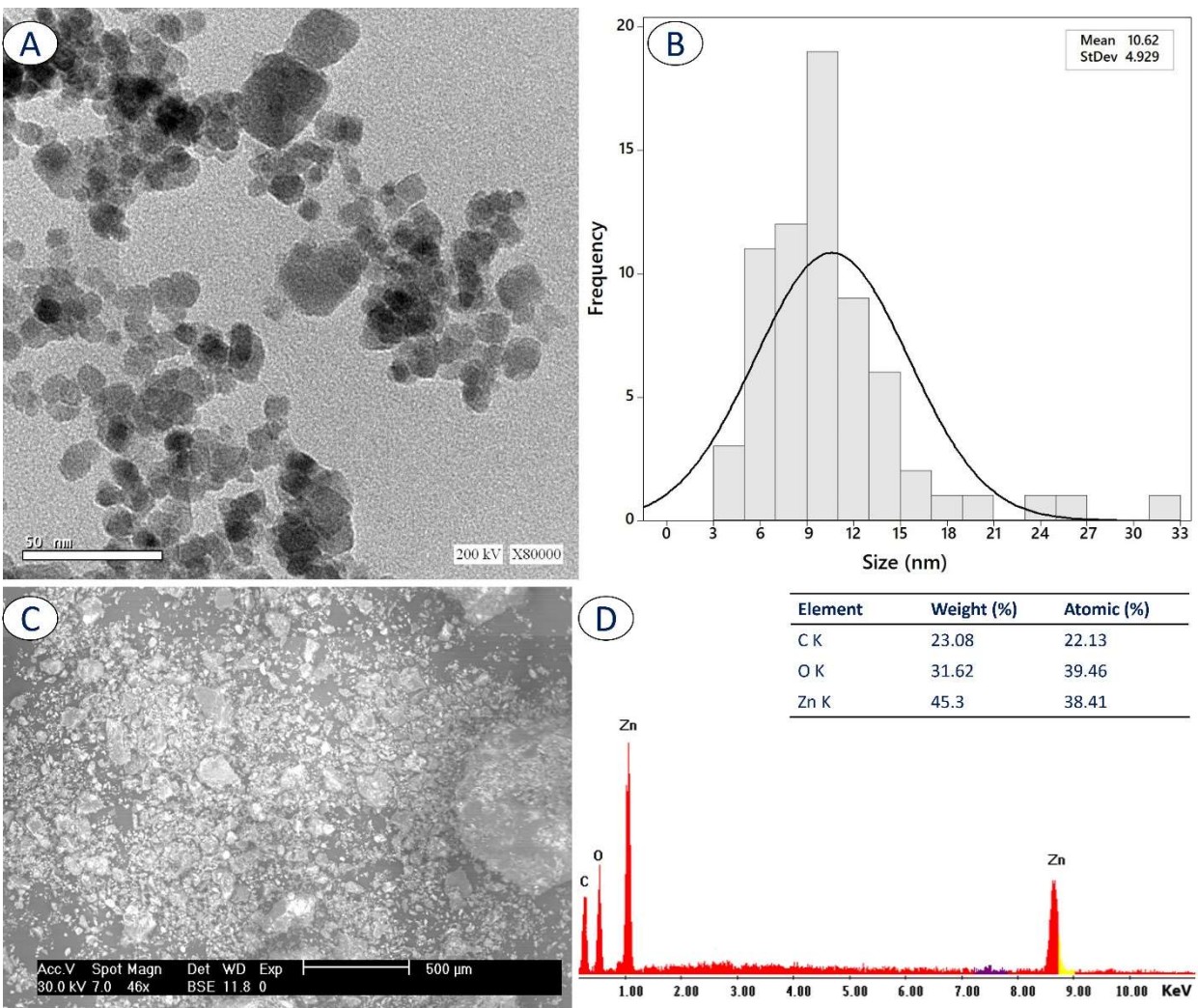

**Figure 3.** Characterization of phyco-synthesized ZnO-NPs: (**A**) is TEM image, (**B**) is size distribution according to the TEM image, (**C**) is SEM image, and (**D**) is EDX analysis.

### 2.2.4. Scanning Electron Microscopy–Energy-Dispersive X-ray (SEM-EDX)

Figure 3C shows the SEM analysis of phyco-synthesized ZnO-NPs that exhibit a state of well dispersion with a spherical shape. The qualitative and quantitative composition

of as-formed ZnO-NPs were assessed using EDX analysis (Figure 3D). The presence of an absorption peak of O at 0.5 Kev and peaks of Zn at 1.0 and 8.68 KeV confirmed the presence of ZnO in the prepared sample [45]. Data showed that the sample contains C, O, and Zn with weight and atomic percentages of (23.08, 31.62, and 45.3) and (22.13, 39.46, and 38.41), respectively. The EDX chart confirmed the high purity of the prepared sample due to the absence of extra peaks, and this finding was confirmed by the XRD analysis. The presence of C may be attributed to capping agents from an algal extract such as polysaccharides, carbohydrates, or proteins [21]. The obtained data are in complete agreement with the published data which reported that the main components of ZnO-NPs fabricated with the aqueous extract of *Nostoc* sp. were Zn and O with weight percentages of 31.8 and 63.2%, respectively [50]. Moreover, the ZnO-NPs fabricated by *S. marginatum* and *U. lactuca* mainly contained Zn and O with weight percentages of (51.6 and 48.4%) and (48.3 and 51.7%), respectively [21].

### 2.2.5. X-ray Diffraction (XRD)

The crystalline nature of phyco-synthesized ZnO-NPs was investigated using the XRD analysis (Figure 4). As shown, there are seven diffraction peaks at two theta (2 θ) values of 31.44°, 34.32°, 36.24°, 47.39°, 56.5°, 62.8°, and 67.12°, which correspond to crystal planes of (100), (002), (101), (102), (110), (103), and (201), respectively (Figure 4). The obtained XRD results are indexed to the hexagonal wurtzite crystalline nature according to the Joint Committee on Powder Diffraction Standards (JCPDS) card number 89-1397. The obtained data are compatible with those recorded by Ishwarya et al. [28] and Azizi et al. [35]. Hamouda et al. reported that the strength and sharp diffraction peaks in the XRD indicate the high crystallinity of synthesized ZnO-NPs [51].

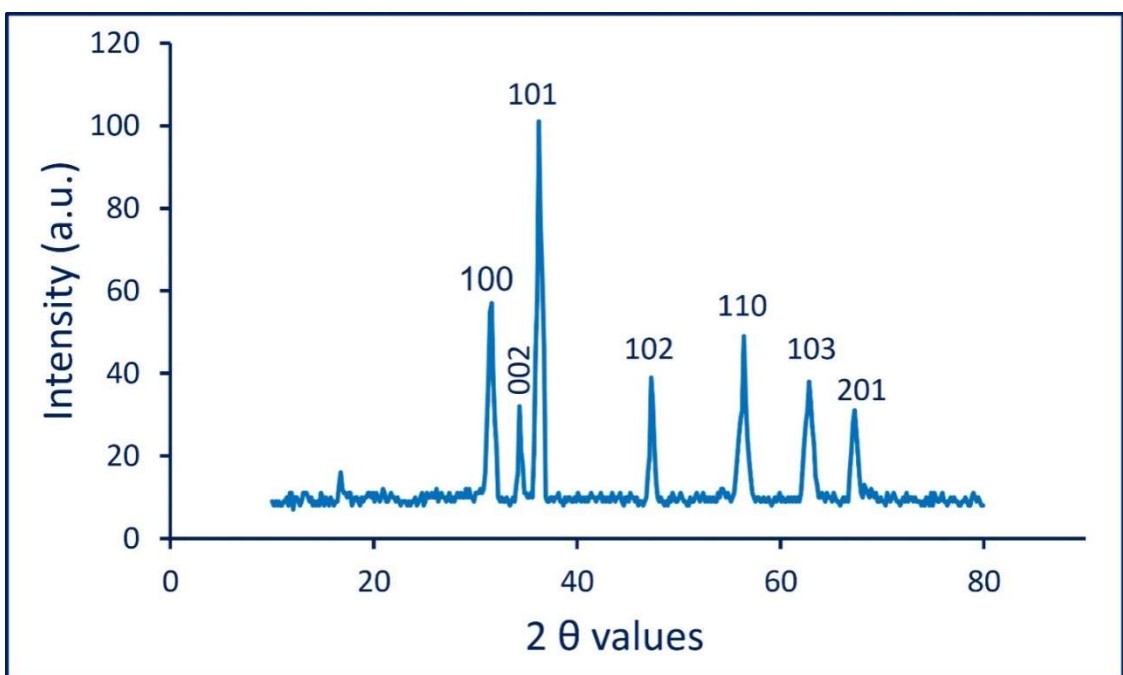

**Figure 4.** X-ray diffraction of ZnO-NPs that were synthesized by the aqueous extract of *U. fasciata*.

The XRD analysis is used to calculate the crystallite size, which gives the possibility to identify or calculate the smaller NPs sizes [52]. The average crystallite size of our synthesized ZnO-NPs was calculated using Scherrer's equation:

$$D = \frac{0.89 \times 1.54}{\beta \ \cos \ \theta} \tag{3}$$

where D is the average ZnO-NPs size, 0.89 is Scherrer's constant, 1.54 is the wavelength of X-ray, β is half of the maximum intensity, and θ is Bragg's diffraction angle. The calculation of the mean crystallite size corresponds to the maximum intense peak (101) at a 2θ value of 36.24° as reported previously [53]. The data analysis revealed that the mean crystallite size of our phyco-synthesized ZnO-NPs was around 28 nm. The obtained data are compatible with the previous reports that indicated that the average crystallite size of ZnO-NPs that were synthesized by leaf aqueous extract of *Tabernaemontana divaricata* was 36.8 nm [22].

### 2.3. Antibacterial Activity

The antibacterial effects of the NPs that have been fabricated by numerous strains of micro- and macroalgae receive great attention due to the huge active metabolites of these organisms. Herein, the synthesized ZnO-NPs by the *U. fasciata* strain showed high antibacterial activity against Gram-positive and Gram-negative bacteria. The data analysis showed that the antibacterial-effect intensity was dependent on the concentration of the used dose of the ZnO-NPs (Figure 5). The highest activity was observed at the concentration of 200 μg mL$^{-1}$ against *Pseudomonas aeruginosa* followed by *Escherichia coli*, *Bacillus subtilis*, and *Staphylococcus aureus* with inhibition zones of 21.7 ± 0.6, 18.7 ± 0.6, 14.7 ± 0.6, and 14.7 ± 1.2 mm, respectively (Figure 5). The activity was decreased by lowering the ZnO-NPs concentration. At 100 μg mL$^{-1}$, the zone of inhibition was 20.3 ± 0.6, 16.3 ± 0.5, 11.8 ± 0.3, and 12.3 ± 0.5 mm against *P. aeruginosa*, *E. coli*, *B. subtilis*, and *S. aureus*, respectively. This observation is compatible with many published studies that refer to the dose-dependent antibacterial activity of ZnO-NPs that were fabricated by various biological entities. For instance, the maximum inhibition of the bacterial growth of *S. aureus, Streptococcus pyogenes*, and *E. coli* were achieved after treatment with 100 μg mL$^{-1}$ ZnO-NPs, whereas the minimum inhibition percentages were obtained at 20 μg mL$^{-1}$ [31]. The antimicrobial activity of ZnO-NPs that were prepared by the brown algae, *Cystoseira crinita*, against *Bacillus cereus, Salmonella typhi, S. aureus, E. coli, Aspergillus niger*, and *Candida albicans* was increased at the high concentration of (10 mg mL$^{-1}$) and decreased gradually at the concentration range of 1000 to 39.06 μg mL$^{-1}$ [54].

The minimum ZnO-NPs concentration that can inhibit the growth of the pathogenic bacterial strains was defined as the minimum inhibitory concentration (MIC). It is important to measure the MIC values of different bioactive compounds against different pathogenic microbes. In the current study, the activity of gradual concentrations of ZnO-NPs (100, 50, 25, 12.5, 6.25, and 3.125 μg mL$^{-1}$) were investigated against the pathogenic Gram-positive and Gram-negative bacteria to identify the MIC value for each strain. The data analysis showed that the MIC values were 6.25 and 25 μg mL$^{-1}$ for *P. aeruginosa* and *E. coli* with inhibition zones of 11.3 ± 0.5 and 10.7 ± 0.5 mm, respectively. Whereas the MIC values for the Gram-positive bacteria *B. subtilis* and *S. aureus* were 50 and 25 μg mL$^{-1}$, respectively, with inhibition zones of 10.7 ± 0.6 and 9.3 ± 0.5 mm (Figure 5). The activity of nanomaterials depends on different parameters such as shape, size, surface area, charge, and Zeta potential. For instance, the antimicrobial activity of the nanorods of ZnO-NPs against *P. aeruginosa, E. coli, B. subtilis* and *S. aureus* was more effective when compared to that of the hexagonal shape [33]. Moreover, the nanopyramids-ZnO exhibited high antibacterial activity against the pathogenic *E. coli* compared to that of nanoplates and nanospheres [55]. The size of the nanomaterial is a critical factor affecting their activity. There is an inverse relationship between the size of NPs and their activity, i.e., the activity was increased as the size decreased [8]. For example, the nanorod-ZnO exhibited a strong antibacterial activity against *E. coli* and *S. aureus* as compared to the microparticles; authors attributed this result to the high surface area [56] due to the small size of the phyco-synthesized ZnO-NPs.

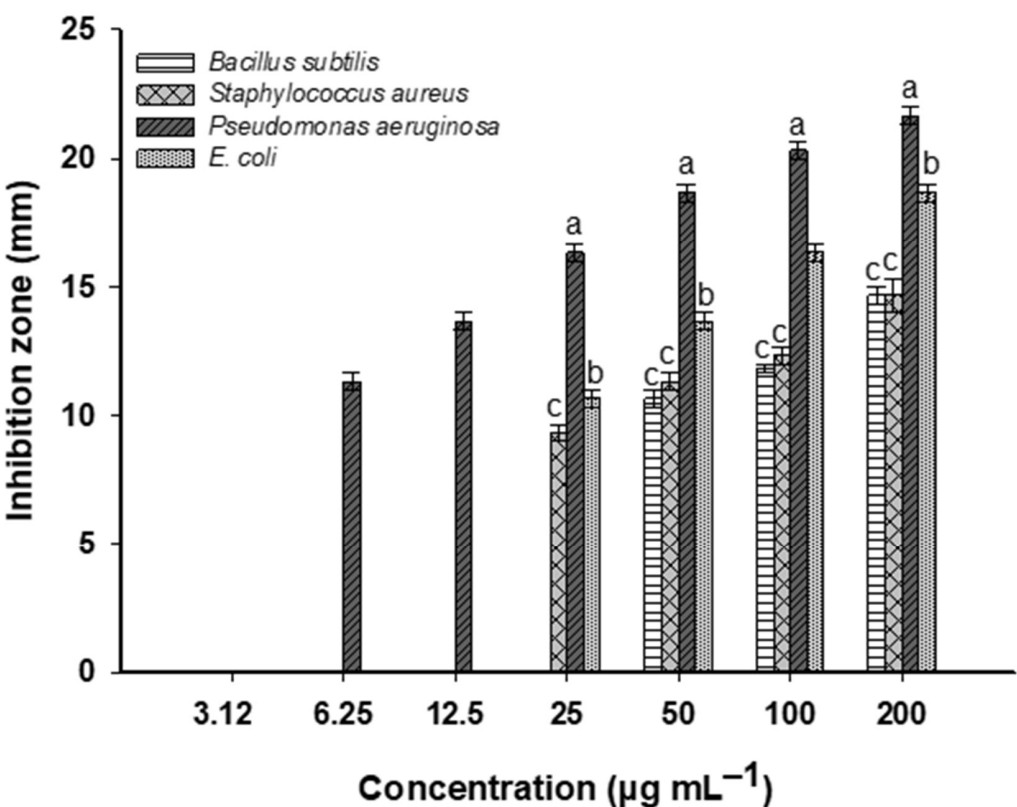

**Figure 5.** Antibacterial activity of *U. fasciata*-mediated green synthesis of ZnO-NPs against the pathogenic Gram-positive (*B. subtilis* and *S. aureus*) and Gram-negative bacteria (*P. aeruginosa* and *E. coli*). The different letters at the same concentration refers to the statistically significant differences ($p \leq 0.05$, $n = 3$, error bars are ±SD).

In the current study, the ZnO-NPs exhibited stronger activity against Gram-negative bacteria than Gram-positive bacteria. This finding can be explained by the different biochemical structure of the cell wall of both bacterial types, which in Gram-positive bacteria is composed of a thick layer of peptidoglycan followed by an internal phospholipids layer (cytoplasmic membrane). These thick peptidoglycan layers may retard the penetration of ZnO-NPs [57]. In contrast, the cell wall of the Gram-negative bacteria contains a thin layer of peptidoglycan followed by high amounts of lipopolysaccharides (LPS). The inhibitory effects of NPs might be attributed to the electrostatic attraction between the negative charge of the LPS and the positively charged NPs [46,58].

The antimicrobial activity of ZnO-NPs can be attributed to three different mechanisms, including interrupting the permeability of cytoplasmic membrane, the release of toxic ions ($Zn^{2+}$) inside the bacterial cell, and/or enhancing the reactive oxygen species (ROS). Due to the direct interaction between bacterial cells and ZnO-NPs, the selective permeability of the cytoplasmic membrane could be changed due to altering the channels and transporter proteins that are overlapping with the cell membrane [59]. Once ZnO-NPs enter the bacterial cell, toxic ions including $Zn^{2+}$ are liberated due to the dissolution of ZnO-NPs inside the cells. These toxic ions might negatively affect metabolism active sites of enzymes, amino acids synthesis, and/or the proton motive force. Ultimately, the active transport system could be stopped. All of these dysfunctions can lead to bacterial cell death [60,61]. As reported previously, the production of toxic ions is shape- and size-dependent on the NPs. For example, the maximum productivity of $Zn^{2+}$ and $Cu^{2+}$ ions due to the dissolution of the ZnO-NPs and CuO-NPs inside the bacterial cells were noticed for the spherical shape and the small-size NPs compared to the rod-shaped and larger particles, respectively. This could be due to the increased solubility equilibrium of smaller and spherical shapes [62]. The accumulation of these toxic ions inside the cells can enhance the oxidative stress and

hence generate ROS. The inhibitory effect of the ROS is due to their oxidizing properties that can damage the bacterial cellular components, i.e., DNA, lipids, proteins, amino acids, and ribosomes [63,64].

### 2.4. Photocatalytic Activity

The catalytic degradation of the MB using nanocatalyst ZnO-NPs was investigated under dark and visible-light irradiation conditions. The degradation of the MB was assessed at interval times of 20, 40, 60, 80, 100, 120, and 140 min in the presence of various nanocatalyst concentrations of 0.25, 0.5, 0.75, and 1.0 mg mL$^{-1}$. The photocatalytic activity of a nanocatalyst is influenced by various factors such as surface morphology, bandgaps, sizes, crystallinity, shape, and concentration [65,66]. The data analysis showed that the degradation of the MB either in light or dark incubation conditions was dependent on the contact time between the catalyst and the dye as well as on the concentration of the ZnO-NPs (Figure 6). At the concentration of 0.25 mg mL$^{-1}$ in dark conditions, the decomposition percentages of the MB were $17.8 \pm 1.8\%$ after 20 min and increased to $43.1 \pm 4.01\%$ after 140 min. In contrast, the decomposition percentages of the control were $4.7 \pm 1.5\%$ after 20 min and increased to $11.8 \pm 2.9\%$ after 140 min. (Figure 6). In the presence of visible-light irradiation, the degradation percentages were $38.1 \pm 0.7$, $47.6 \pm 1.1$, and $49.4 \pm 0.8\%$ after 20, 120, and 140 min, respectively, in the presence of 0.25 mg mL$^{-1}$ of the catalyst as compared to the control ($4.9 \pm 1.7$, $10.4 \pm 3.1$, and $13.3 \pm 2.9\%$ after 20, 120, and 140 min, respectively). These results could be due to the existence of more active sites on the surface of the catalyst when the concentration of the ZnO-NPs was increased. The latter may enhance the sorption of dye and increase the ability to form radicals that may improve the degradation efficiency [67,68].

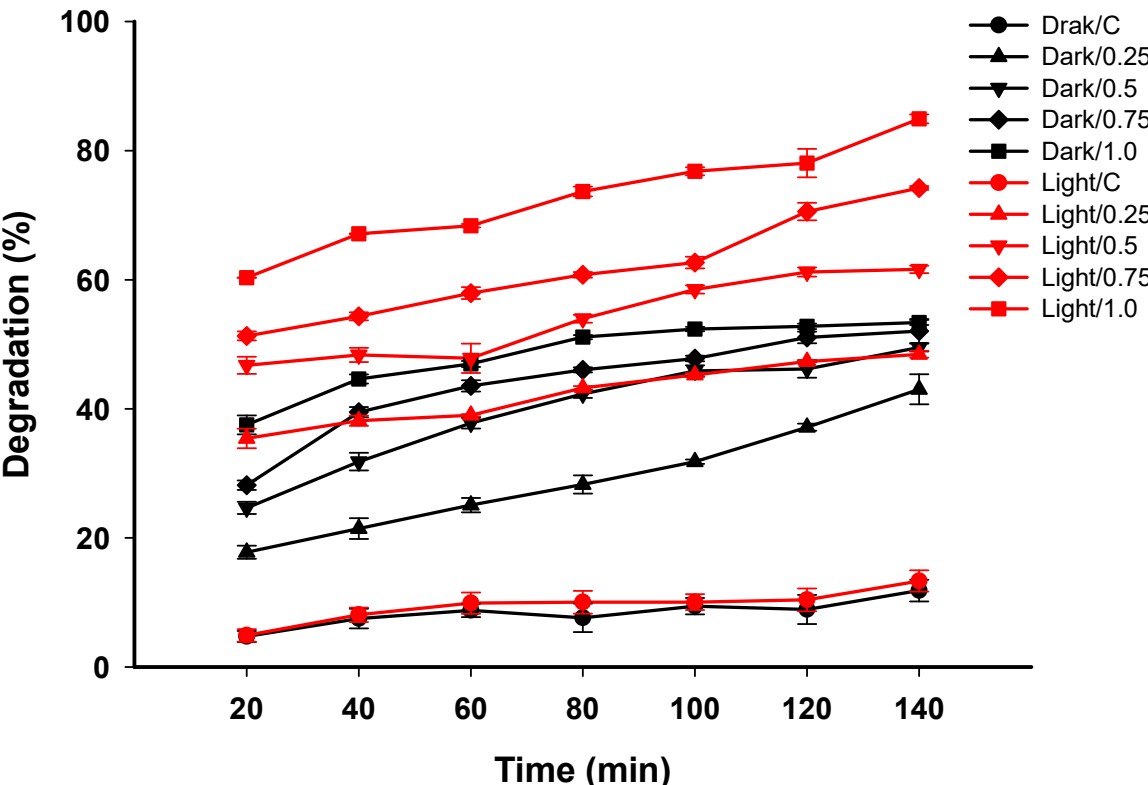

**Figure 6.** Degradation efficacy of methylene blue at different ZnO-NPs concentrations (0.25, 0.5, 0.75, and 1.0 mg mL$^{-1}$) after interval times (20, 40, 60, 80, 100, 120, and 140 h) under dark and visible light irradiation conditions.

In the current study, the degradation efficacy of the MB was increased at high concentrations of the ZnO-NPs. Therefore, the degradation percentages of the MB were increased up to $49.5 \pm 1.1\%$ (in dark) and $61.6 \pm 0.9\%$ (in light irradiation) after 140 min in the presence of $0.5 \text{ mg mL}^{-1}$ of the nanocatalyst (Figure 6). The analysis of variance revealed no significant differences between the decomposition values of the MB in the presence of light irradiations and $0.5 \text{ mg mL}^{-1}$ of the catalyst after 120 ($61.3 \pm 1.2\%$) and 140 min ($61.6 \pm 0.9\%$). The nanocatalyst ZnO fabricated by the marine alga, *Turbinaria ornata*, showed an efficacy to decompose approximately 38% of an aqueous solution containing rhodamine-B dye after 90 min in sunlight [69]. Kahsay et al. reported that the concentration of a catalyst has a significant effect on its photocatalytic efficiency [70]. The obtained data are compatible with this previous finding; the highest MB degradation was achieved at a concentration of $1.0 \text{ mg mL}^{-1}$. The data analysis showed that the degradation percentage of $53.4 \pm 0.7\%$ was obtained after 140 min in the presence of $1.0 \text{ mg mL}^{-1}$ of the nanocatalyst under dark conditions. This value was increased up to $84.9 \pm 1.2\%$ under the visible-light irradiation (Figure 6). Various published studies reported that the photodegradation efficacy of nanomaterials increases with an increase in the irradiation time [42,69,71]. The obtained data are in agreement with those that reported that the maximum photocatalytic degradation of methylene blue, rhodamine-B, and orange-II by ZnO-NPs synthesized by an aqueous extract of *Dolichos lablab* were obtained at a catalyst concentration of $1.0 \text{ g L}^{-1}$, followed by 0.75 and $0.6 \text{ g L}^{-1}$ [70].

Recently, various nanoparticles were used to decolorize and treat real wastewater containing different dyes. For instance, Ag-NPs synthesized by the *B. marisflavi* strain TF-11 showed a photocatalytic degradation with percentages of 96.9% for wastewater containing different azo dyes, including Methyl Red, Direct Blue-1, and Reactive Black-5. Moreover, the physicochemical characteristics of wastewater including electrical conductivity, pH, sulfates, chlorides, TDS, TSS, COD, and BOD were highly improved [72]. In our recent study, MgO-NPs synthesized by the fungal strain, *Aspergillus niger* F1, exhibited high efficacy in the treatment of real textile wastewater under light irradiation conditions. In addition, MgO-NPs exhibited efficacy in the treatment of tanning wastewater and the removal of various heavy metals such as Pb, Co, Cd, Cr, and Ni with percentages of $72.7 \pm 1.3$, $63.4 \pm 1.7$, $74.1 \pm 1.8$, $94.2 \pm 1.2$, and $70.8 \pm 1.5\%$, respectively [26]. The CuO-NPs also showed high activity in relation to removing colors and COD from the textile wastewater. The maximum decolorization (83%) was achieved at optimum conditions of 0.05 g/L catalyst concentration and a pH value of 6.9 for 60 min in the presence of UV-irradiation conditions. Whereas the maximum COD reduction (99%) was achieved at a pH of 6.8 and CuO-NPs concentration of 0.02 g/L for 60 min under the UV-irradiation conditions [73].

In the current study, the maximum decolorization was obtained in the presence of visible-light irradiation. This phenomenon is attributed to the photoexcitation, separation, and hence migration of charges from the valence band (VB) to the conductance band (CB) of the nano-ZnO catalyst, and finally the surface oxidation–reduction mechanism [69,74,75]. Under light irradiation conditions, the surface of the catalyst (ZnO) is hit by photonic energy (hv) that is equal to or greater than the band gap of ZnO. As a result, electrons are excited to form electron–hole pairs; one of them ($h^+$) is transferred to the VB and the other ($e^-$) is transferred to the CB, Equation (4). The generated electron pairs are then transferred to the surface of the catalyst to be involved in redox reactions. The $h^+$ on the VB interacts with the $H_2O$ or OH (hydroxide ions) forming $^\bullet OH$ (hydroxyl radicals), Equation (5). Moreover, the $e^-$ on the CB are reacting with $O_2$ forming superoxide anion radical ($^\bullet O_2^-$) followed by the formation of hydrogen peroxide ($H_2O_2$), Equations (6)–(8). Finally, the generated radical species will react with the MB dye and are converted to generate intermediate compounds that will be hydrolyzed to green compounds (i.e., $CO_2$ and $H_2O$), Equation (9) (Figure 7) [26,76]. The disappearance of the dye color in dark conditions could be due to the sorption of the dye over the surface of the ZnO-NPs. The main feature of the nanomaterials is the size of the surface area; the smaller sizes have a larger surface area

and hence increase the adsorption sites. In the current study, the small size of the ZnO-NPs (3–33 nm) indicates the successful sorption of the MB on the surface. This mechanism is efficient to explain the removal of the MB using ZnO-NPs in dark conditions as reported previously [4].

$$ZnO \xrightarrow{h\nu} h^+_{VB} + e^-_{CB} \tag{4}$$

$$h^+ + H_2O \text{ or } OH^- \rightarrow {}^\bullet OH + H^+ \tag{5}$$

$$e^- + O_2 \rightarrow {}^\bullet O_2{}^- \tag{6}$$

$${}^\bullet O_2{}^- + H^+ \rightarrow HO_2{}^\bullet \tag{7}$$

$$HO_2{}^\bullet + HO_2{}^\bullet \rightarrow H_2O_2 + O_2 \tag{8}$$

$$MB + \text{generated reactive species} \rightarrow \text{intermediate compounds} \rightarrow CO_2 + H_2O \tag{9}$$

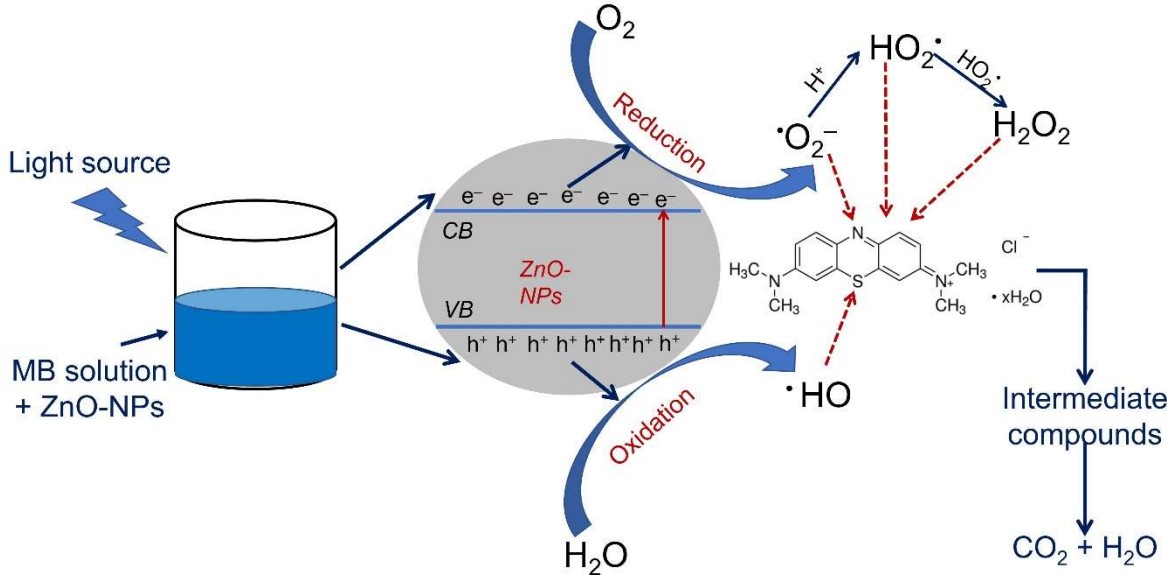

**Figure 7.** Prospective model for the photocatalytic mechanism of ZnO-NPs-mediated degradation of MB dye.

The reusability of a catalyst for several cycles with almost the same efficacy is receiving great attention in catalytic studies. Therefore, the reusability potential of ZnO-NPs for the degradation of the MB dye has been investigated for six consecutive cycles under the optimum conditions (in the presence of 1.0 mg mL$^{-1}$ under the visible-light irradiations, and the degradation percentages were assessed after 140 min). The catalyst was collected from the first cycle by centrifugation, washed thrice with distilled H$_2$O, and dried before being reused in the second cycle. The obtained data showed that the activity of the ZnO-NPs decreased by a percentage of 5.2% after six consecutive cycles. The dye color removal percentages were 83.3 ± 3.7, 82.5 ± 1.3, 81.3 ± 1.9, 80 ± 2.7, 79.9 ± 2.2, and 79 ± 0.9% for first, second, third, fourth, fifth, and sixth cycle, respectively (Figure 8). This modest decrease in the decolorization potential might be attributed to the gradual overload or saturation of the adsorption sites on the ZnO-NPs' surface by some intermediate compounds of the initial MB dye decomposition event. [77]. Moreover, the minimal decrease in the photocatalytic activity can be related to the unavoidable losses of the catalyst during the consecutive cycles [78]. In a recent study, the catalytic efficacy of ZnO-NPs that were formulated by the *Penicillium corylophilum* strain As-1 to degrade the MB dye was decreased from 98 to 94% with a reduction percentage of 4% after ten consecutive batches [74]. Moreover, the efficacy of ZnO-NPs synthesized by the peeling aqueous extract of the *Passifora foetida* to decompose the MB dye was decreased from 93.3

to 86.6% after four cycles with a reduction percentage of 7.2% [79]. The slight reduction in the capacity of ZNO-NPs as a catalyst after several successive cycles, in the current study, is confirming the stability of the ZnO-NPs. Hence, it is possible to use our product in different biotechnological applications.

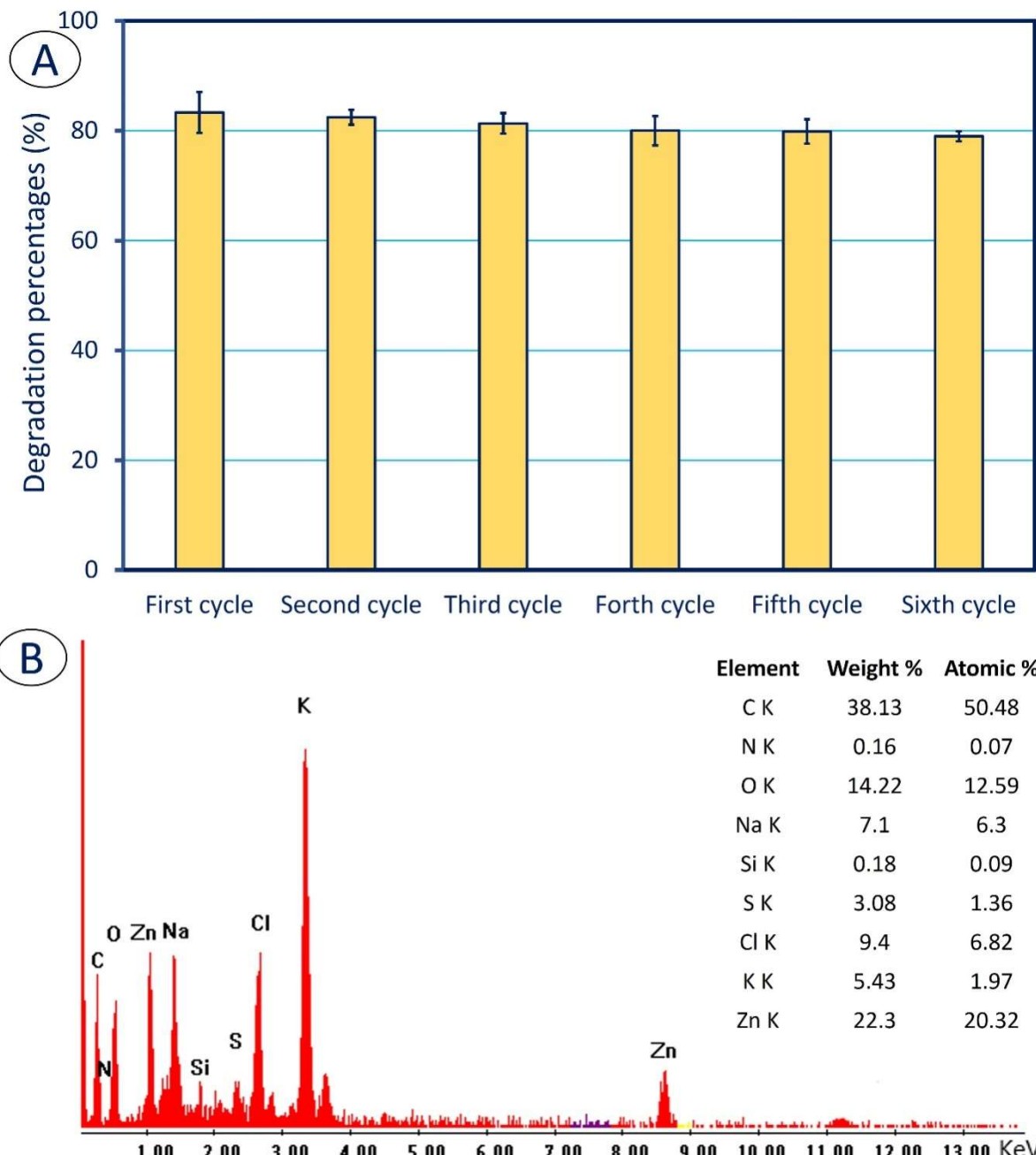

**Figure 8.** (**A**) Investigates the stability of ZnO-NPs during the degradation of MB dye for different consecutive cycles. (**B**) The EDX analysis of the ZnO-NPs collected from the sixth cycle to investigate the elementary mapping after MB dye adsorption.

The qualitative and quantitative elementary mapping of the nanocatalyst-ZnO was analyzed for the sample that was collected from the sixth cycle to confirm the successful adsorption of the MB dye and confirm the activity of the catalyst. The appearance of new peaks, as well as the decrease in the intensity of the main peaks, indicates the adsorption of

the MB dye. As shown in Figure 8B, the appearance of peaks of N, S, and Cl with weight and atomic percentages of (0.16, 3.08, and 9.4%) and (0.07, 1,36, and 6.82%), respectively, can be related to the adsorption of the MB dye on the catalyst surface. Moreover, there is an increase in the weight and atomic percentages of the C peak from 23.08 and 22.13%, respectively, in the original sample to 38.13 and 50.48% due to the decomposition of the MB dye. The decrease in the weight and atomic percentages of Zn in the sample from 45.3 and 38.41%, respectively, to 22.3 and 20.32% might be attributed to the gradual loss of the catalyst during the consecutive cycles.

### 2.5. Treatment of Tanning Wastewater

The activity of phyco-synthesized ZnO-NPs in the treatment of tanning wastewater was investigated at the optimum conditions of the catalytic activity which were 1.0 mg mL$^{-1}$ of the catalyst in the presence of visible-light irradiations for 140 min. The greenish-blue color of tanning wastewater is due to the use of chromium at high concentrations [80]. In the current study, the results of the physicochemical parameters of tanning wastewater after treatment indicate the efficacy of phyco-synthesized ZnO-NPs in significantly reducing all parameters ($p < 0.01$) when compared to the untreated effluent. At the end of the photocatalytic reaction, the synthesized ZnO-NPs exhibited high efficacy in removing tanning effluent color with a percentage of 96.1 $\pm$ 1.7% (Table 1). The physicochemical parameters, including salinity, chemical oxygen demand (COD), biological oxygen demand (BOD), total dissolved solids (TDS), total suspended solids (TSS), conductivity, and heavy metals, are represented at high values in tanning wastewater and cannot be discharged without treatment [81,82]. These high values are due to the random usage of multi-component chemicals, such as bicarbonates, sulfates, chlorides, calcium, phosphates, nitrates, potassium, and sodium, in the tanning industry [83]. Fortunately, the phyco-synthesized ZnO-NPs can reduce these parameters with percentages of 89, 89, 97, and 92% for the COD, BOD, TSS, and conductivity, respectively (Table 1). These high removal percentages might be attributed to the use of a high dose of ZnO-NPs (1.0 mg mL$^{-1}$) in the presence of light which led to an increase in the amount of adsorption active sites and enhancement of the generation of active radical species.

**Table 1.** Physicochemical characterization of tanning wastewater before and after ZnO-NPs treatment to investigate their efficacy to remove Cr (VI) ions.

| Parameter | Unit | Control | After Treatment with ZnO-NPs | Removal Percentages (%) |
|---|---|---|---|---|
| pH | | 6.3 $\pm$ 1.1 | 8.1 $\pm$ 0.2 | - |
| Decolorization | % | 3 $\pm$ 0.3 | 96 $\pm$ 2 | |
| COD | mg L$^{-1}$ | 1017 $\pm$ 5 | 114 $\pm$ 4 | 89 |
| BOD | mg L$^{-1}$ | 2305 $\pm$ 7 | 264 $\pm$ 6 | 89 |
| TSS | mg L$^{-1}$ | 8847 $\pm$ 7 | 268 $\pm$ 7 | 97 |
| Conductivity | S·m$^{-1}$ | 27,363 $\pm$ 6 | 2337 $\pm$ 5 | 92 |
| Cr (VI) | mg L$^{-1}$ | 864 $\pm$ 6 | 57 $\pm$ 4 | 93 |

Although the leather industry has an economic value in many countries, tanneries are considered one of the most significant sources of heavy metals accumulation in the ecosystem. Chromium (Cr), as chromium sulfate, is the main heavy metal in tanning processes that may cause severe damage to the soil and the underground water if discharged without proper treatment [84]. In addition, chromium may be accumulated in food chains, causing animal and human disorders, such as lung cancer and skin allergies [24]. ZnO-NPs have a high efficacy to adsorb various heavy metal ions from the wastewater as reported previously [85]. In the current study, the chromium ion in tanning effluent was decreased from 867 $\pm$ 6 mg L$^{-1}$ (control) to 57 $\pm$ 4 mg L$^{-1}$ with removal percentages of 93% after the treatment with ZnO-NPs under the determined optimum conditions (Table 1). These high removal percentages might be attributed to the small size of ZnO-NPs that led to an increase

in the surface area and, hence, an increase in the adsorption sites on the surface of the adsorbent. The high removal percentages could be attributed to the reduction/oxidation of metal ions by electron pairs ($e^-_{CB}$ and $h^+_{VB}$) that are generated due to exposure to light. The metal ion reduction was achieved when the redox potential of metal is more positive than the $e^-_{CB}$ of the ZnO particle, whereas the metal oxidation was achieved when its oxidation potential is less positive than $h^+_{VB}$ [86]. Similarly, green-synthesized ZnO-NPs had the efficacy to remove Pb(II) from an aqueous solution with a percentage of 93% [87]. Pandey et al. reported that approximately 26.7 mg g$^{-1}$ of Cr (VI) was removed using ZnO-NPs from an aqueous solution with an initial chromium concentration of 30 mg L$^{-1}$ [88].

### 2.6. Comparative Study

Due to the unique properties of ZnO-NPs, they are synthesized using various micro- and macroalgae to be incorporated into various applications (Table 2). These activities are shape- and size-dependent. For instance, an aqueous extract of *U. lactuca* and *S. marginatum* was used to form a round and spherical shape with sizes of 12–17 and 6–11 nm, respectively, that display varied antimicrobial and anticancer activity [21]. Whereas the sponge-asymmetrical shape of ZnO-NPs with a size range of 10–50 nm was fabricated using the aqueous extract of *U. lactuca* and showed high activity in removing the MB dye in the presence of sunlight. In addition, it exhibits a high antibacterial activity against Gram-positive and Gram-negative bacteria as well as larvicidal activity against *Aedes aegypti* [28]. In the current study, the synthesized ZnO-NPs exhibited high activity due to their small size (3–33 nm). It also showed a high efficacy to inhibit the growth of the pathogenic bacteria *S. aureus*, *B. subtilis*, *E. coli*, and *P aeruginosa* at low concentrations. In addition, it showed a high potential to remove the MB dye, improve the physicochemical properties of tanning wastewater, and demonstrate high adsorption of Cr (VI).

**Table 2.** Comparing size, shape, and activities of ZnO-NPs synthesized by various micro- and macroalgal species.

| ZnO-NPs Synthesized by | Shape | Size | Applications | Reference |
|---|---|---|---|---|
| *Ulva lactuca* *Stoechospermum marginatum* | Round shape and spherical | 12–17 nm 6–11 nm | Antimicrobial and in vitro cytotoxicity | [21] |
| *Cystoseira crinita* | multilayered rectangular shape | 23–200 nm | Antimicrobial and antioxidant activity | [54] |
| *Chlorella* sp. | Spherical | 20–50 nm | Photocatalytic activity | [39] |
| *Arthrospira platensis* | Spherical | 30–55 nm | Antimicrobial and in vitro cytotoxicity | [18] |
| *Sargassum muticum* | Hexagonal | 30–57 nm | - | [35] |
| *Anabaena cylindrica* | Rod shape | 40–60 nm | Antimicrobial and in vitro cytotoxicity | [89] |
| *Hypnea musciformis* | Spherical | 29–35 nm | Antibacterial activity and ecotoxicity against *Artemia salina* | [29] |
| *Nostoc* sp. EA03 | Star shape | 50–80 nm | Antibacterial and in vitro cutotoxicity | [50] |
| *Ulva lactuca* | Sponge-asymmetrical shape | 10–50 nm | Photocatalysis, antibiofilm, and larvicidal activity | [28] |
| *Sargassum wightii* | Spherical shape | 40–50 nm | Antibiofilm and larvicidal activity | [32] |
| *Ulva fasciata* | Spherical | 3–33 nm | Antibacterial, photocatalytic efficacy, tanning wastewater treatment, and heavy metal (Cr (VI)) removal | Current study |

## 3. Materials and Methods

### 3.1. Algae Collection

The marine green macroalgae, *Ulva fasciata* Delile, Ulvophyceae, Chlorophyta, was collected from the beach of Ras at Tin, Qasim Al Gomrok, Alexandria Governorate, Egypt

(GPS: N31°12′37.4976″, E29°52′54.2856″). The collected algae were transferred directly to the Phycology Laboratory, Al-Azhar University, Cairo, Egypt, and underwent identification based on the standard key of Taylor [90]

### 3.2. Preparation of the Algal Aqueous Extract

The collected marine seaweed, *U. fasciata* Delile, was rinsed thrice with distilled $H_2O$ and cut into small pieces before being left to air dry. After complete dryness, the algal pieces were grinded into a fine powder. Approximately 10 g of seaweed powder was mixed with 100 mL of dis. $H_2O$ and subjected to boiling for 30 min under stirring conditions. After that, the boiling extract was centrifuged at 10,000 rpm to collect the supernatant which was then used as a reducing agent for $Zn(CH_3COO)_2 \cdot 2H_2O$ [28].

### 3.3. Seaweed Aqueous Extract-Mediated Green Synthesis of ZnO-NPs

Approximately 10 mL of the seaweed aqueous extract was added to 90 mL of zinc acetate solution (44 mg was dissolved in 90 mL dis $H_2O$) under stirring conditions for 60 min at 50 °C to obtain a final concentration of 2 mM. After that, the previous mixture was left at room temperature for 24 h in dark. The obtained white precipitate was collected and washed thrice with dist. $H_2O$ and subjected to oven-dry at 300 °C for 2 h before using [18].

### 3.4. Characterizations

The color of algal aqueous extract and the as-formed ZnO-NPs was monitored by measuring the absorbance at a wavelength range of 200–600 nm to detect the maximum surface plasmon resonance (SPR) of ZnO-NPs. Approximately 2 mL of overnight ZnO-NPs solution was added to the cuvette and measured their absorbance using the spectrophotometer (JENWAY 6305, Staffordshire, UK). The functional groups assigned to different algal metabolites in aqueous extract and their ability to be used as a reducing and capping agent for ZnO-NPs were investigated using Fourier transform infrared (FT-IR) spectroscopy. Approximately 10 mg of ZnO-NPs powder was mixed well with KBr to form a disk before scanning at a range of 400–4000 $cm^{-1}$ (Agilent system Cary 660 FT-IR model, Tokyo, Japan) [91]. The size and shape of algal-mediated green-synthesized ZnO-NPs were assessed using Transmission Electron Microscopy (TEM) (JEOL 1010, Tokyo, Japan, 200 kV). A few drops of ZnO-NPs colloidal solution were deposited over the TEM grid and underwent vacuum desiccation overnight before being analyzed. The surface morphology and elemental composition of the sample were analyzed by Scanning Electron Microscopy connected with energy-dispersive X-ray (SEM-EDX) (JEOL, JSM-6360LA, Tokyo, Japan) [14]. X-ray diffraction (XRD) (Malvern Panalytical X′PERT PRO MPD, Philips, Eindhoven, Netherlands) was used to analyze the crystallinity nature of ZnO-NPs. The XRD analysis was achieved at two theta scales of 10°–80°. The analysis conditions were voltage, 40 KV; current, 30 mA; and Cu Ka was used as an X-ray radiation source.

### 3.5. Antibacterial Activity

The activity of ZnO-NPs to inhibit the growth of the pathogenic bacteria *Staphylococcus aureus* ATCC6538, *Bacillus subtilis* ATCC6633 (Gram-positive bacteria), *Escherichia coli* ATCC8739, and *Pseudomonas aeruginosa* ATCC9022 (Gram-negative bacteria) was assessed by the agar-well diffusion method [92]. Each bacterial strain was inoculated in nutrient broth media and incubated overnight at 37 ± 2 °C. At the end of the incubation period, the optical density (O.D.) for each strain was adjusted at 1.0. A 50 μL from each bacteria solution was pipetted onto the center of the sterilized petri dish and mixed well with cooled sterile Muller Hinton agar media. Upon media solidification, wells (0.6 cm in diameter) were made in each plate and filled with 100 μL of the prepared ZnO-NPs solution (200 μg mL$^{-1}$) and transferred to the refrigerator for one hour before being incubated at 37 ± 2 °C for 24 h. The diameter of the clear zone around each well (mm) was measured. The experiment was performed in triplicate.

### 3.6. Determination of the MIC Value

The highest concentration (200 μg mL$^{-1}$) of the synthesized ZnO-NPs exhibited antimicrobial activity against all tested microbes. Therefore, various concentrations (100, 50, 25, 12.5, 6.25, and 3.125 μg mL$^{-1}$) were prepared to detect the MIC (minimum inhibitory concentration) value using the agar-well diffusion method as mentioned above. Approximately 50 μL from each bacterial-strain solution was pipetted on a sterilized petri dish followed by a cooled Muller Hinton agar media over inoculum and mixed well. After that, 100 μL of each prepared concentration was added to wells (0.6 mm) prepared in the solidified plate before being kept in the refrigerator for one hour. The loaded plates were incubated for 24 h at 37 ± 2 °C. The MIC value was recorded as the lowest concentration that has the efficacy to inhibit microbial growth [93].

### 3.7. Photocatalytic Activity

The efficacy of algal-mediated green synthesis ZnO-NPs in the degradation of methylene blue (MB) as a model dye was investigated. The catalytic experiment was conducted in both dark and visible-light (100-Watt, Halogen tungsten lamp with λ > 420 nm and a light intensity of 2.87 W m$^{-2}$, the distance between the light source and MB solution was 30 cm) irradiation using 10 mg L$^{-1}$ of MB dye solution. The pH value and temperature of the MB solution were adjusted at 7 and 35 °C, respectively. The concentrations of the used ZnO-NPs were 0.25, 0.5, 0.75, and 1.0 mg mL$^{-1}$. Before the experiment, the MB dye solution containing the nanocatalyst was subjected to stirring for 30 min to confirm the absorption/desorption equilibrium of the dye on the nanocatalyst surface [28]. After that, the MB dye solution containing the nanocatalyst was incubated either in dark or light at 35 °C and pH 7 under aeration condition. During the experiment, 2 mL of the treatment was withdrawn at regular interval times (20, 40, 60, 80, 100, 120, and 140 min) and subjected to centrifugation at 5000 rpm for 10 min. The control was running with the experiment under the same conditions (stirring in the presence or absence of light irradiation) in absence of nanocatalyst. The optical density of the clear supernatant was measured using the M-ETCAL spectrophotometer at maximum MB $\lambda_{max}$ (664 nm). The degradation percentages were calculated using the following equation [42].

$$\text{Degradation percentages } (\%) = \frac{\text{Absorbance at zero time} - \text{absorbance at interval time}}{\text{Absorbance at zero time}} \times 100 \tag{10}$$

The reusability of the nanocatalyst was investigated for six successive cycles. The catalyst was collected from the first cycle by centrifugation and rinsed thrice with dis. H$_2$O to remove any adhering particles followed by drying in the oven at 50 °C before being used in the next cycle.

### 3.8. Treatment of Tanning Wastewater

The green-synthesized ZnO-NPs were used to investigate their efficacy in the treatment of tanning wastewater under the determined optimum conditions of the catalytic experiment. The experiment was carried out in 250 mL conical flasks containing 100 mL tanning wastewater mixed with the optimum concentration of ZnO-NPs under light irradiation conditions. The mixture was stirred for 30 min to confirm the absorption/desorption equilibrium. The color removal of tanning wastewater due to treatment with ZnO-NPs was measured using the above-mentioned equation.

At the end of the experiment, the biological oxygen demand (BOD), chemical oxygen demand (COD), total suspended solids (TSS), total dissolved solids (TDS), and conductivity as indicators for the successful treatment were assessed according to the standard protocols [94].

Because chromium (Cr) is the most common heavy metal in tanning wastewater, it was quantified before and after the ZnO-NPs treatment by using the atomic adsorption spectroscopy (A PerkinElmer Analyst 800 atomic spectrometer).

*3.9. Statistical Analysis*

Data represented in the current study are the means of three independent replicates. Data were analyzed using the statistical package SPSS v17. The mean difference comparison between the treatments was analyzed by the *t*-test or the analysis of variance (ANOVA) and subsequently by the Tukey HSD test at $p < 0.05$.

## 4. Conclusions

The metabolites of the aqueous extract of the macroalgae *U. fasciata* have the ability to mediate the biosynthesis of ZnO-NPs. The physicochemical characterization of the phyco-synthesized ZnO-NPs was performed by UV–vis spectroscopy, FT-IR, TEM, SEM-EDX, and XRD. Data showed that the color of the algal aqueous extract was changed to white after mixing with the metal precursor, indicating the formation of ZnO-NPs that exhibited a maximum surface plasmon resonance at 330 nm. Moreover, FT-IR revealed the role of the algal metabolites in the aqueous extract in capping and stabilizing the newly formed NPs. TEM and XRD indicated the formation of spherical-shape NPs with a size range of 3–33 nm with a crystalline nature. The SEM-EDX chart confirms the presence of Zn and O in the sample with weights of 45.3 and 31.62%, respectively. The activity of these NPs has been proven through its significant antibacterial and catalytic activity. The antibacterial activity was dose-dependent and had the efficacy to inhibit the growth of the pathogenic bacterial strains at different concentrations with varied clear zones. The catalytic activity to remove the MB dye was better in the presence of visible light at a concentration of 1.0 mg mL$^{-1}$ after 140 min at pH 7 and 35 °C. The maximum dye removal was 84.9 $\pm$ 1.2% compared to 53.4 $\pm$ 0.7% of the dark conditions. The phyco-synthesized ZnO-NPs showed high activity to decrease the chemical parameters of tanning effluent with percentages of 88.8, 88.5, 96.9, and 91.5% for the COD, BOD, TSS, and conductivity, respectively. Moreover, the maximum Cr (VI) removal was achieved after the treatment of the tanning effluent with ZnO-NPs with a percentage of 93.4%. We proudly can recommend the usage of our product to potentially control the growth of pathogenic bacteria and prevent the possible environmental pollution with the dyes and heavy metals.

**Author Contributions:** Conceptualization, A.F. and A.M.E.; methodology, A.F., A.M.E., A.A., H.A.S., E.F.E.-B. and S.E.-D.H.; software, A.F., A.A., E.F.E.-B., D.H.M.A., K.S.A. and S.E.-D.H.; validation, A.F., A.M.E., A.A.,D.H.M.A., K.S.A. and H.A.S.; formal analysis, A.F., A.A., H.A.S., E.F.E.-B. and S.E.-D.H.; investigation, A.F., A.M.E. and S.E.-D.H.; resources, A.A., H.A.S., D.H.M.A., K.S.A. and E.F.E.-B.; data curation, A.F., A.M.E., H.A.S. and S.E.-D.H.; writing—original draft preparation, A.F., H.A.S., E.F.E.-B. and S.E.-D.H.; writing—review and editing, A.F. and A.A.; visualization, A.F., A.M.E., H.A.S. and A.A.; supervision, A.F. and S.E.-D.H.; project administration, A.F.; funding acquisition, H.A.S., D.H.M.A. and K.S.A. All authors have read and agreed to the published version of the manuscript.

**Funding:** This research received no external funding.

**Acknowledgments:** Authors extend their appreciation to Botany and Microbiology Department, Faculty of Science, Al-Azhar University, Cairo, Egypt; and Botany Department, Faculty of Science, Fayoum University for the great cooperation and supporting to achieve and publication of this research work. Also, authors thank Princess Nourahbint Abdulrahman University Researchers Supporting Project number PNURSP2022R15, Princess Nourahbint Abdulrahman University, Riyadh, Saudi Arabia. As well, our thank is extended to Imam Mohammed Bin Saud Islamic University (IMSIU), Riyadh, Saudi Arabia for supporting the publication of this research work.

**Conflicts of Interest:** The authors declare no conflict of interest.

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
