# Peer review of "Phyco-Synthesized Zinc Oxide Nanoparticles Using Marine Macroalgae, Ulva fasciata Delile, Characterization, Antibacterial Activity, Photocatalysis, and Tanning Wastewater Treatment"

_catalysts, doi:10.3390/catal12070756_

Round 1

Reviewer 1 Report

The current work is well carried out but authors should try to emphasize better the importance of current manuscript in order to attract the readership of the journal. Before consideration of this manuscript for publication in this journal the authors should address the following major points. 

1. The introduction of this manuscript is too casual. Authors should improve the introduction of the manuscript with the description of novelty of this work along with scientific explanation as compared as recent studies published in this research area.

2. It will be better if authors will provide a comparison table in order to summarize characteristic features of the similar particle study in literature.

3. The yield of prepared nanoparticles along with description of nanoparticle synthesis cost must be provided in the revised manuscript.

4. Authors should elaborate the key components of macroalgae which are useful in nanoparticles synthesis and its quantification in specific weight of macroalgae. 

5. Authors should explore the utility of the developed nanoparticles in real dye contaminated water from industries, at least 2-3 sample studies. 

6. English of the manuscript should be thoroughly  checked and corrected.

7. The mechanism of the nanoparticles synthesis, antimicrobial activity and photocatalytic activity needs substantial improvements with scientific explanation and instrumental proofs. 

Author Response

Dear reviewers thank you very much for your valuable comments. We answered all comments point by point as shown in the revised manuscript and author response. Please see the attachment.

Reviewer 2 Report

Phyco-Synthesized Zinc Oxide Nanoparticles Using Marine Macroalgae, Ulva fasciata Delile, Characterization, Antibacterial Activity, Photocatalysis, and Tanning Wastewater Treatment.

The manuscript A. Fouda et al. about antibacterial activity, photocatalysis properties, and tanning wastewater treatment for series of ZnO catalysts is well written and interesting and the authors wide characterized synthetized materials i.a. diffraction and electron microscope techniques. The results of the researches are useful, the experiments are carefully conducted and the paper is well written. Therefore, I recommend the acceptance of the manuscript after minor revision.

I have two technical comments regarding the manuscript

1. The text formatting should be corrected, for example the equation (line 126-129) contains numerous errors, for example the dot in the hydrate is at the bottom instead of the middle of the line in the following equations there are unnecessary dots, for example 2h. instead of 2h

2. Equation 9 needs improvement. For example, in the OH radical there is no indicated electron, only empty fields (squares). Please correct it.

I also have two comments regarding the experimental and discussion part of manuscript.

After reading the article carefully, I have a question about chapters 2.4 and 2.5. The authors of the article described the purification water from MB by photodegradation in the presence of UV light with a wavelength of λ = 420 nm and a strong decrease in the content of Cr (VI) ions in water. In six consecutive attempts of photocatalytic decomposition of MB, a slight decrease in the activity of ZnO-NPs was observed.

1. In the case of subsequent MB photodegradation cycles for the same catalyst batch, I suggest that the Zn (II) content be determined for each cycle (Fig. 8). Preference can be given to using the ICP-MS or ICP-OES technique. ZnO is amphoteric and therefore insoluble in water, however the pH of MB is not neutral. This may affect the leaching of ZnO into the solution and cause the phenomenon of quasi-homogeneous catalysis. Please comment obtain results in chapter 2.4.

2. Due to the fact that Zn (II) compounds are harmful to aquatic organisms, I suggest that the authors in Table 1 add information about Zn (II) content before and after ZnO-NPs treatment. The authors determined the content of Cr (IV) by atomic absorption spectroscopy (AAS), if possible, I suggest to use the same techniques to determine the zinc content or to use the ICP-MS or ICP-OES technique.

In general, after clarifying my questions I recommend to publish this manuscript in MDPI Catalysts.

Author Response

Dear reviewer, thank you very much for your valuable comments. We answered all comments point by point as shown in the revised manuscript and author response. Please see the attachment.

Round 2

Reviewer 1 Report

Authors revised manuscript as per suggested points. This form of manuscript can be accepted for publication. However, I am not satisfied with reply of my cooment no. 3  about yield of Nanoparticles by this adopted approach and comment no. 4 about identification of key components of macroalgae which are useful in nanoparticles synthesis and its quantification in specific weight of macroalgae.

Author Response

Thank you very much for your approval